# Surface coupling in Bi$_2$Se$_3$ ultrathin films by screened Coulomb interaction

Jia-nan Liu[1,2], Xu Yang[1,3], Haopu Xue[1,2], Xue-song Gai[1], Rui Sun[1,2], Yang Li[1,2], Zi-Zhao Gong[1,2], Na Li[1,2], Zong-Kai Xie[1,2], Wei He[1], Xiang-Qun Zhang[1], Desheng Xue[4] & Zhao-Hua Cheng [1,2,3] ✉

Single-particle band theory has been very successful in describing the band structure of topological insulators. However, with decreasing thickness of topological insulator thin films, single-particle band theory is insufficient to explain their band structures and transport properties due to the existence of top and bottom surface-state coupling. Here, we reconstruct this coupling with an equivalently screened Coulomb interaction in Bi$_2$Se$_3$ ultrathin films. The thickness-dependent position of the Dirac point and the magnitude of the mass gap are discussed in terms of the Hartree approximation and the self-consistent gap equation. We find that for thicknesses below 6 quintuple layers, the magnitude of the mass gap is in good agreement with the experimental results. Our work provides a more accurate means of describing and predicting the behaviour of quasi-particles in ultrathin topological insulator films and stacked topological systems.

Topological insulators (TIs) are known for their surface states protected by time-reversal symmetry[1]. Owing to the strong spin-orbit coupling (SOC) and non-zero $\mathbb{Z}_2$ number[1], the Kramers' pairs of these topological surface states exhibit quasi-particle behaviors with different dispersion relations, such as Dirac fermions, type-I, and type-II Weyl fermions, as well as corresponding spin-momentum locking[2]. Many properties induced by the surface states have been investigated extensively such as quantum spin Hall (QSH) effect[3], quantum anomalous Hall (QAH) effect with magnetic doping[4], and nonlocal magnetization dynamics in ferromagnet/TI heterojunction[5]. These topological surface states are generally housed within a few nanometers of the surfaces of samples, while the bulk samples are usually not ideal insulators owing to the existence of vacancies during the sample preparation. Ultrathin TI films with a large surface-to-volume ratio are ideal systems to amplify the dominance of surface states. Meanwhile, with decreasing thickness of ultrathin films, the coupling of top and bottom surface states increases. As a typical example of a $\mathbb{Z}_2$ topological insulator, the topological surface state of Bi$_2$Se$_3$ with a Dirac fermion dispersion relation

$E_\pm(\mathbf{k}) = \mu \pm \sqrt{(\hbar v_F)^2 k^2 + m^2}$ was predicted by the single-particle band theory with the help of the tight binding approximation and the model Hamiltonian, where $\mu$ is the position of the Dirac point, $v_F$ is the Fermi velocity, $\hbar$ is the reduced Plank constant and $m$ is the mass of the Dirac fermion[6–8]. This dispersion relation was experimentally observed[9]. For the Bi$_2$Se$_3$ samples with thickness above 6 quintuple layers (QLs), the dispersion of surface state is gapless, i.e., $m = 0$, while the coupling between top and bottom surface states can open an energy gap for the samples with thickness of below 6 QL, i.e. $m \neq 0$. It should be noted that, within the framework of single-particle band theory, the surface states are treated as a two-level system described by the massive Dirac Hamiltonian $H_D = \hbar v_F (\boldsymbol{\sigma} \times \mathbf{k}) \cdot \hat{\mathbf{z}} + m\sigma_z + \mu$. The Chern number of each energy band in such systems, which is proportional to Hall conductivity by the TKNN formula[10], depends on the sign of mass gap $m$[1,11].

In previous works on Bi$_2$Se$_3$ ultrathin films[11,12], the single-particle band theory predicated that with decreasing thickness, a non-zero mass gap $m$ induced by the surface states coupling appears and evolves from a

[1]State Key Laboratory of Magnetism and Beijing National Laboratory for Condensed Matter Physics, Institute of Physics Chinese Academy of Sciences, Beijing 100190, China. [2]School of Physical Sciences, University of Chinese Academy of Sciences, Beijing 100049, China. [3]Songshan Lake Materials Laboratory, Dongguan, Guangdong 523808, China. [4]Key Laboratory for Magnetism and Magnetic Materials of the Ministry of Education, Lanzhou University, 730000 Lanzhou, China. ✉e-mail: zhcheng@iphy.ac.cn

negative to positive value around 2.5 QL. This change in the sign of $m$ will induce a topological phase transformation, and a non-monotonic evolution of the energy gap. However, this non-monotonic change in energy gap was not observed by the angle-resolved photoemission spectroscopy (ARPES) measurements, and the observed energy gaps are much larger than the theoretical ones[13]. To further investigate this non-monotonic change, In-doped $Bi_2Se_3$ films are studied and it was found that the topological phase transformation does not appear below 6 QL, which confirms the trivial topological nature of the system[14]. Moreover, on the basis of single-particle band theory, a non-zero $m$ will cause a spin polarization $\sigma_z$ in the z-direction near the $\Gamma$ point[11], while this spin polarization was not observed in the spin-resolved ARPES measurement[15]. The nano-scaled multi-tip transport experiments also confirm that the 1D ballistic edge state of a 2D topological QSH insulator is absent in TI ultrathin films[16], implying that the surface states are topologically trivial. Meanwhile, both transport experiments[17,18] and numerical calculation[19] suggest that TIs in the 2D limit are intrinsically quantum many-body systems. It seems that the single-particle band theory is not sufficient to describe the band structure and transport properties in ultrathin $Bi_2Se_3$ TI films.

In this work, we investigate the coupling between the top and bottom surface states in the form of a screened Coulomb potential. Both the band shifting and the mass gap are analyzed in terms of the Hartree self-energy and the self-consistent gap equation, respectively. The band shifting and the mass gap of $Bi_2Se_3$ films with different thicknesses are extracted from ARPES spectra and fit the theoretical results well. Within the framework of our theory, the energy gap may originate from the breaking of chiral symmetry instead of the time-inversion symmetry for one band of a Kramers' pair, which is similar to the case in graphene with EEI[20,21]. Therefore, the energy gaps in the ultrathin $Bi_2Se_3$ films with thickness of below 6 QL are always topologically trivial and the z-component of spin polarization shall not appear, which is consistent with reported experimental results. Compared with the single-particle band theory, our work provides a more accurate means to describe and predict the behavior of quasi-particles, especially in TI ultrathin films and/or stacked systems.

## Results

The Hamiltonian of massless Dirac fermions is $H = \hbar v_F \boldsymbol{\alpha} \cdot \mathbf{k} - \mu_0$, where $\alpha_i = \gamma_i \gamma_0$ ($i = 1$ and $2$) are the base vectors of the spin space and $\gamma_\mu$ are the gamma matrices. Similar to the EEI in graphene[21], the whole Lagrangian of the top ($\alpha = +1$) and bottom ($\alpha = -1$) surface states $\psi_\alpha$ in TIs, as shown in Fig. 1a, is $L = L_0 + L_{int}$, where

$$L_0 = \hbar v_F \sum_\alpha \int d^2\mathbf{r}\, \bar{\psi}_\alpha(t,\mathbf{r}) i\boldsymbol{\gamma} \cdot \nabla \bar{\psi}_\alpha(t,\mathbf{r}), \tag{1}$$

$$L_{int} = \int d^2\mathbf{r} d^2\mathbf{r}' \sum_{\alpha,\alpha'} \bar{\psi}_\alpha(t,\mathbf{r})\gamma_0\psi_\alpha(t,\mathbf{r}) V_{\alpha,\alpha'}(t,\mathbf{r};t',\mathbf{r}';d)\bar{\psi}_{\alpha'}(t',\mathbf{r}')\gamma_0\psi_{\alpha'}(t',\mathbf{r}'), \tag{2}$$

$\mathbf{r}$ and $\mathbf{r}'$ are the in-plane 2-dimensional position vectors, $d$ is the distance between the top and bottom surfaces, i.e., the thickness of samples. $V_{\alpha,\alpha'}(t,\mathbf{r};t',\mathbf{r}';d) = \delta_{\alpha,-\alpha'}V(t,\mathbf{r};t',\mathbf{r}';d)$ describes the coupling between these two surface states. Since the quasi-particles are essentially electrons, the dominant interaction is the Coulomb interaction. The Fermi velocity, $v_F$, of Dirac fermions in Dirac material is about $\sim 10^5$ m/s[13], which is much slower than the speed of light. The screened Coulomb interaction between top and bottom fermions is

$$V(t,\mathbf{r};t',\mathbf{r}';d) = \delta(t-t')V(|\mathbf{r}-\mathbf{r}'|;d)$$
$$= \delta(t-t')\frac{e^2}{4\pi\varepsilon_0}\frac{\exp\left[-\lambda^{-1}\sqrt{|\mathbf{r}-\mathbf{r}'|^2+d^2}\right]}{\sqrt{|\mathbf{r}-\mathbf{r}'|^2+d^2}}, \tag{3}$$

where $e$ is the electron charge, $\varepsilon_0$ is vacuum permittivity. $\lambda$ is the screening length, $\sqrt{|\mathbf{r}-\mathbf{r}'|^2+d^2}$ is the distance between the top and bottom electron with position vector $\mathbf{r}$ and $\mathbf{r}'$, and $\delta(t-t')$ is the delta function. Since it is difficult to calculate the two-dimensional Fourier transformation of $V(r,d)$ with respect to $\mathbf{r}$, we adopt an approximate expression that reads

$$V(q,d) := \mathscr{F}_\mathbf{r}\left[V(r,d)\right] \approx \frac{e^2}{4\pi\varepsilon_0}\frac{\exp\left[-d\sqrt{q^2+\lambda^{-2}}\right]}{\sqrt{q^2+\lambda^{-2}}} \tag{4}$$

To analyze the error induced by this approximation, we numerically calculated the $\mathscr{F}_\mathbf{r}\left[V(r,d)\right]$ as $V_{num}(q,d)$ and compare it with the expression $V(q,d)$. The relative error $\left|\frac{V_{num}-V}{V_{num}}\right|$ is less than $10^{-5}$ (see Supplementary Note 1). The free propagator of $L_0$ is $S_0(k_0,\mathbf{k}) = (\gamma_0 k_0 - \hbar v_F \boldsymbol{\gamma} \cdot \mathbf{k})^{-1}$. After turning Wick rotation and considering the Fermi surface shifting owing to vacancy doping present in the fabricated samples, the complete propagator modified by $L_{int}$ is $S(k_0,\mathbf{k}) = \left[\gamma_0(ik_n - \mu)A_1(k) - \hbar v_F \boldsymbol{\gamma} \cdot \mathbf{k}A_2(k) - m(k)\right]^{-1}$, where $m(k)$ is the renormalization mass gap. $A_{1,2}$ was denoted as the wave-function renormalization function[21]. $ik_n = i(2n+1)\pi/\beta$ is the Matsubara frequency, $\beta = 1/k_B T$ and $k_B$ is the Boltzmann constant. It should be noted that the mass term caused by interaction does not relate to the spin polarization, which naturally makes the none of spin in the z-direction hold. This result is based on the fact that the Coulomb interaction, rather than the time-inversion symmetry, breaks the chiral symmetry[22]. In this work, we focus on the calculation of the first two quantities $\mu$ and $m$. So far, only three parameters are unknown, the Fermi velocity $v_F$ in $L_0$, the Fermi energy $\mu_0$, and the screening length $\lambda$ in $L_{int}$. The former two parameters can be extracted experimentally by ARPES spectra in a thick enough sample. The last one describes the surface state decay length, which is approximately equal to half of the critical thickness when the energy gap opens[23], can be calculated by the density functional theory numerically.

Because of the $\delta_{\alpha,-\alpha'}$ term in $L_{int}$, it is easy to check that only the Hartree term survives in all one-loop self-energy, which is

$$\Sigma_{H\alpha} = V(q=0,d)\rho_{-\alpha}, \tag{5}$$

where $V(q=0,d) = \frac{e^2}{4\pi\varepsilon_0}\lambda\exp\left[-d/\lambda\right]$ is the DC part of $V(q,d)$ and $\rho_{-\alpha}$ is the areal density of fermions on the opposite surface. With the Dyson equation $S^{-1} = S_0^{-1} + \Sigma_H$, $\Sigma_H$ makes the whole energy band shift since it is a real number. The second order of $L_{int}$ gives the self-consisted gap equation for $m(ik_n,\mathbf{k})$. Since we are only concerned with the mass gap at the $\Gamma$ point, following by the widely used approximation to solve interaction-induced mass gap problems in the Dirac system, we set $A_{1,2} = 1$[21,22,24,25] and treat $m(k)$ as a constant[22,25,26]. This treatment allows us to take the trace of $m$, and consequently obtain the self-consistent gap equation as follows

$$m(ik_n,\mathbf{k}) = \frac{1}{N_\gamma}\text{Tr}\frac{1}{\beta}\sum_m\int\frac{d^2\mathbf{q}}{(2\pi)^2}V^2(q,d)\gamma_0 S_\sigma(ik_n - iq_m,\mathbf{k}-\mathbf{q})\Pi_{-\sigma}(iq_m,\mathbf{q})\gamma_0, \tag{6}$$

where $N_\gamma$ is the rank of chosen gamma matrices, $\Pi_{-\sigma}(iq_m,\mathbf{q})$ is the polarization function of the Dirac fermion in the $-\sigma$ surface defined as

$$\Pi_{-\sigma}(iq_m,\mathbf{q}) = -\frac{2}{\beta}\sum_l\int\frac{d^2\mathbf{p}}{(2\pi)^2}\gamma_0 S_{-\sigma}(ip_l,\mathbf{p})\gamma_0 S_{-\sigma}(ip_l + iq_m,\mathbf{p}+\mathbf{q}). \tag{7}$$

The related self-energy is shown as blue lines in Fig. 1a. The polarization function of Dirac fermions has been well investigated in graphene systems[24]. We calculate the polarization function as follows

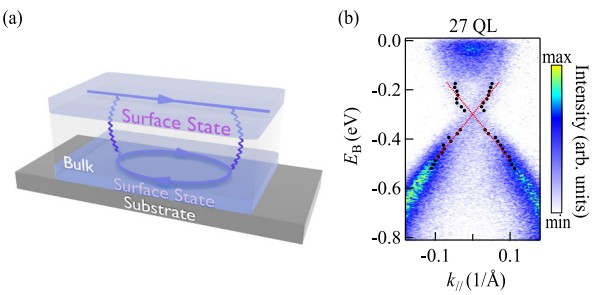

**Fig. 1 | The form of interaction and the parameters extracted from ARPES.
a** Feynman diagram of the self-energy used in the Dyson equation. The wavy lines and arrows represent boson lines and fermion lines, respectively. The polarization function of Dirac fermions is made of the fermion lines on the opposite surface. **b** ARPES spectra of 27 QL $Bi_2Se_3$ sample. The black dots in **b** are the peak positions of the surfaces state fitted by MDCs with the Lorentzian function, and the red dotted line is the linear fitting results of two sets of black dots. $E_B$ is the binding energy and $k_\parallel$ is the in-plane wave vector.

(see Supplementary Note 2).

$$\Pi_\sigma(iq_m,\mathbf{q}) \approx \frac{1}{8\pi\hbar^2 v_F^2}\int_0^1 dx\left[2\varpi_\beta(B) - \frac{C}{B}\frac{\partial}{\partial B}\varpi_\beta(B)\right], \quad (8)$$

where $B = \left[x(1-x)q_m^2 + x(1-x)\hbar^2 v_F^2 q^2 + m^2\right]^{1/2}$, $C = -iq_m\mu_0(1-2x) + 2m^2 + m\hbar v_F\boldsymbol{\gamma}\cdot\mathbf{q} + m[(1-2x)iq_m - 2\mu_0]\gamma_0 + \left[2x(1-x)iq_m + (1-2x)\mu_0\right]\hbar v_F\boldsymbol{\gamma}\cdot\mathbf{q}\gamma_0$, and

$$\varpi_\beta(B) = \frac{1}{\beta}\ln\left[2\cosh\beta B + 2\cosh\beta\mu_0\right] \approx \begin{cases} B & B' > \mu_0 \\ -B & B' < -\mu_0 \\ \mu_0 & -\mu_0 < B' < \mu_0 \end{cases} \quad (9)$$

The approximation holds when $T \to 0$. With decreasing $q$, $V^2(d,q)$ decays at the rate of a Lorentzian function with a width of $\lambda^{-1}$. For $Bi_2Se_3$, the $k$-range of linear Dirac particles is about $0.1\,\text{Å}^{-1}$[17], which is much larger than $\lambda^{-1}\approx 0.035\,\text{Å}^{-1}$. It is therefore reasonable to use the approximation $\int d^2\mathbf{q}V^2(d,q)g(q)\approx\frac{e^4}{64\pi^2\varepsilon^2}\left[\frac{1}{\pi}g(q=0)+\lambda^{-1}\frac{\partial}{\partial q}|_{q=0}g(q)\right]e^{-2\lambda^{-1}d}$ (see Supplementary Note 1) to simplify Eq. (6), where $g(q)$ is a function that does not fluctuate too much, and the definition of the $\delta$ function in polar coordinates is used. Finally, one can obtain the thickness-dependent mass gap (See the Supplementary Note 2 regarding the detailed error analysis) as:

$$m \approx \frac{e^4\mu}{128\pi^3\varepsilon_0^2\hbar^2 v_F^2}\exp\left[-\frac{2d}{\lambda}\right], \quad (10)$$

where $\mu = \mu_0 + \Sigma_H$ described by Eq. (5). In this expression the sign of the mass gap does not change when thickness decreases, and consequently it is consistent with the absence of gap reclosing probed from ARPES.

We choose $Bi_2Se_3$ ultrathin films to investigate the coupling between top and bottom surface states. Eight $Bi_2Se_3$ samples with different thickness of 1–7 QL and 27 QL are grown on Si(111) substrates by molecular beam epitaxy (MBE). The method of sample growth and the measurements of ARPES were described in previous work[27]. Since the interaction term in Eq. (2) decays exponentially with the thickness at the rate of the surface state decay length, $\lambda\approx 3$ QL, it can be ignored for the sample of 27 QL $Bi_2Se_3$, and consequently the dispersion relation of Eq. (1) with $\mu_0$ is $E_\pm(k) = \mu_0 \pm \hbar v_F k$. We extract the parameters $v_F$ in Eq. (1) and the position of the Dirac point $\mu_0$ from the sample of 27 QL as shown in Fig. 1b using the following steps. First, we fit the momentum distribution curves (MDCs) with a Lorentzian function (see

Supplementary Note 3), and the fitted peak positions are marked as black dots in Fig. 1b. Then we linearly fit these two sets of peak positions as shown by red dotted lines in Fig. 1b and obtain $\mu_0\approx -(0.299\pm 0.004)$ eV and $v_F\approx(2.81\pm 0.08)\times 10^5$ m/s. It is noteworthy that there is a difference between $k_F$ and $\mu_0/\hbar v_F$ owing to the quadratic term of $k$. The $k_F$ fitted by MDC at the Fermi level is about $0.9\,\text{Å}^{-1}$, and the areal density of fermions in a single surface is $\rho_\sigma\approx 1.3\times 10^{13}$ cm$^{-2}$. These values are in good agreement with those of the previous work[17]. For the other samples, we used the dispersion relation of the generalized Green function $E_\pm(\mathbf{k}) = \mu \pm \sqrt{\hbar^2 v_F^2 k^2 + m^2}$ due to the existence of energy gap $E_{gap} = 2m$. By fitting the peak positions we can check whether or not there is an energy gap in the energy distribution curve (EDC) at the $\Gamma$ point, and further determine the magnitude of $m$. The remaining steps are the same as those for fitting the sample of 27 QL. More details on peak fitting are present in the Supplementary Note 3. The fitted peak positions and dispersion relation are shown in Fig. 2a–f. The fitted thickness-dependent $\mu$ and $m$ are shown as the scatter plots in Figs. 3a, b, respectively. It should be noticed that, although the Van der Waals materials like $Bi_2Se_3$ grow layer by layer, the film prepared by MBE is still difficult to be atomically flat in the entire area covered by the light spot with the scale of $10^2$ microns in our ARPES measurements[13,28]. The ARPES results should be a superposition of the results between two integer layers. Therefore, to confirm whether the gap will reclose or not at -2.5 QL predicted by the previous theory[11], we further extracted the energy gap of 1.5 and 2.5 QL $Bi_2Se_3$ to improve the measurement accuracy of this thickness-depended evolution near the critical point (more details can be checked in Supplementary Note 3). As shown in Fig. 2b, the non-monotonic evolution of the energy gap around the thickness of 2–3 QL is not found.

We can clearly see that the whole energy band shifts to the deep binding energy with decreasing thickness. In previous works, this phenomenon was attributed to the band bending effect[13,23]. However, the band bending effect at the interface between $Bi_2Se_3$ and vacuum was found to be <0.1 eV for the samples with thickness below 7 QL[29], and it should be much smaller at the interface between $Bi_2Se_3$ and silicon. As shown in Fig. 3a, for the samples with a thickness <6 QL, the band shift is larger than 0.1 eV, therefore cannot be explained by the band bending effect. Meanwhile, as shown in Fig. 3a, the self-energy of the Hartree term, which is described by Eq. (5) with $\rho_\sigma$ and $\lambda$ discussed above, fits the experimental results quite well. The curve described by Eq. (10) is plotted in Fig. 3b. We find that the curve fits both our experimental results and previous work as the cyan circles in Fig. 3b for comparison[13]. Furthermore, we can also explain the reason for none of energy gap reclosing -2–3 QL. It should be mentioned that with the thickness decreasing, the critical thickness of the phase transition from massless to massive Dirac fermion cannot be determined by Eq. (10). Other methods for analyzing critical behavior, such as the bifurcation theory or parameter embedding method[21], are needed to find this critical thickness. In this work, we only consider the magnitude to the energy gap. The energy gap calculated by Eq. (10) at the phase transition thickness is <10 meV, which is smaller than the energy resolution of ARPES or the band broadening in most cases[30]. When the thickness further decreases to 1 QL, the mass gap described by Eq. (10) is about 0.84 eV, which is even larger than the energy gap of the bulk state[6]. In this region, the lower branch of surface state first drops into the bulk states because of the band shifting. As shown in Fig. 4a, the Dirac cone is no longer complete, and consequently, the calculated mass gap by our theory cannot be measured from ARPES for 1 QL.

Only one parameter $\lambda$ is used to describe the coupling between the top and bottom surface by Eq. (2) and Eq. (3), and all thickness-dependent results are based on the form of $\lambda^{-1}d$ in the exponential

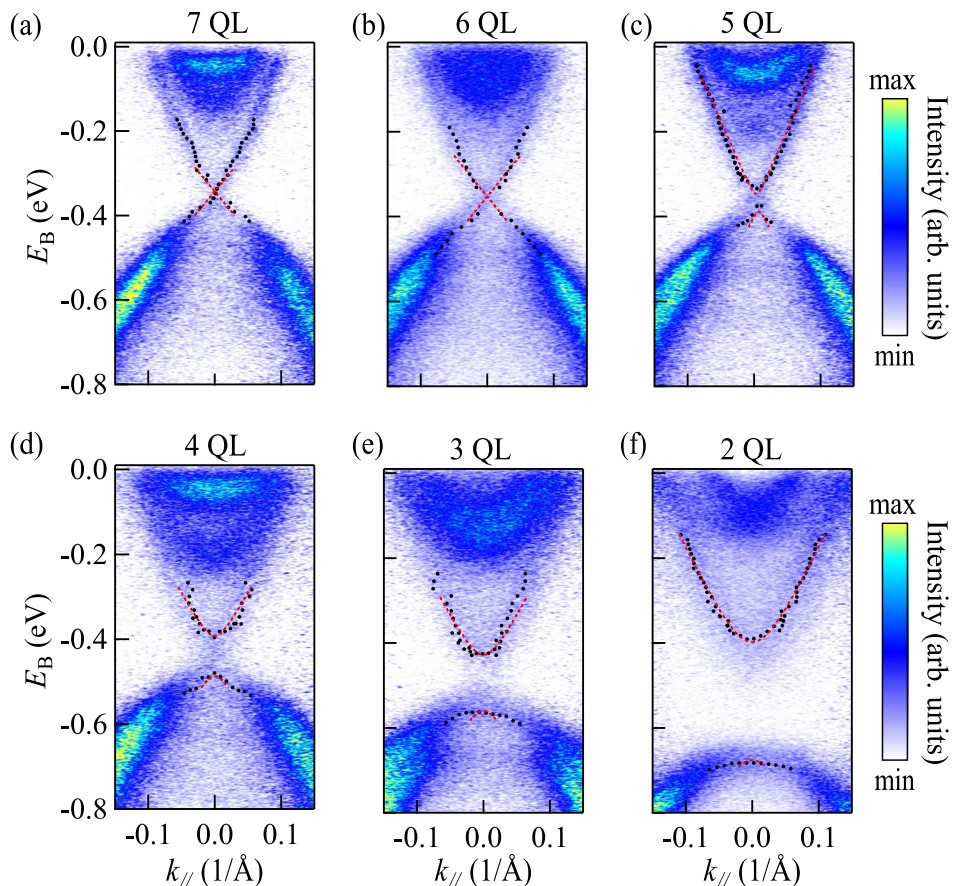

**Fig. 2 | ARPES spectra of Bi$_2$Se$_3$/Si(111) samples with different thicknesses.** **a**–**f** are the ARPES spectra of the Bi$_2$Se$_3$ sample of 7, 6, 5, 4, 3, and 2 QL, respectively. The black dots are the peak positions fitted by MDCs and EDCs, while the red dot lines are the fitting results of black dots with massive or massless Dirac dispersion. $E_B$ is the binding energy and $k_{\parallel}$ is the in-plane wave vector.

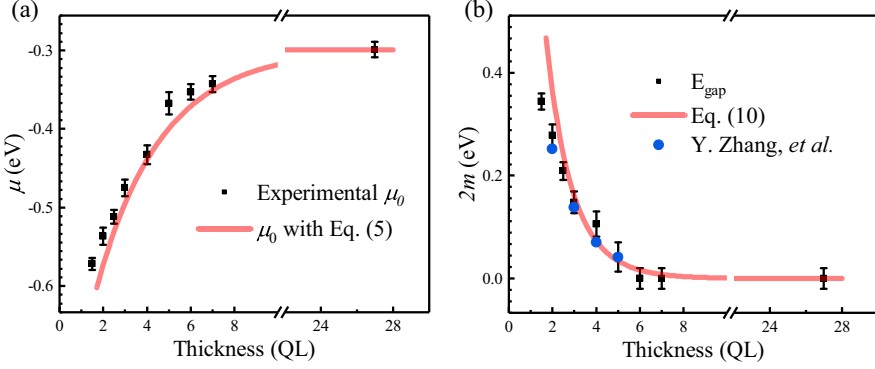

**Fig. 3 | The results of theoretical validation. a**, **b** are the experimental and theoretical results of the Dirac position $\mu$ and the magnitude of the gap $2m$, respectively. The blue dots in **b** are the results of Y. Zhang et al. [13]. The error bars are the root of the sum of the squared peak position fitting error and the squared energy resolution of our ARPES.

function. This method makes the analysis of coupling between surface states intuitive and concise, especially in multilayered systems. For example, we further prepare and measure the ARPES spectra of Bi$_2$Se$_3$(1 QL)/Bi$_2$Te$_3$(2 QL) (noted as Sample S1), and Bi$_2$Se$_3$(1 QL)/Bi$_2$Te$_3$(1 QL)/Bi$_2$Se$_3$(1 QL) (noted as Sample S2) as shown in Figs. 4b and 4c. The screening length of Bi$_2$Te$_3$, $\lambda_{BT} \approx 1$ nm, is less than that of Bi$_2$Se$_3$, $\lambda_{BS} \approx 2.8$ nm[23]. The effective exponential factor $\left[\lambda^{-1}d\right]_{eff} \sim \sum_i \lambda_i^{-1}d_i$ in Sample S1 is about 2.33, which is equivalent to the value found in Bi$_2$Se$_3$ film of about 7 QL, while the effective thickness of Sample S2 is -5 QL

Bi$_2$Se$_3$. One can find the difference between these two samples and the 1 QL of Bi$_2$Se$_3$ as shown in Fig. 4a.

We reconstructed the surface coupling by the screened Coulomb interaction between the top and bottom surface states. The evolution of the energy gap with thickness fits well with the self-cosistent mass gap equation. Interestingly, we find that the Dirac fermion coupling between the top and bottom surface state is quite similar to that between the K and K' points in graphene, in which chiral symmetry breaking with a mass gap is also observed[31]. Considering that the energy gap predicted by the single-particle theory[11] is less than one-

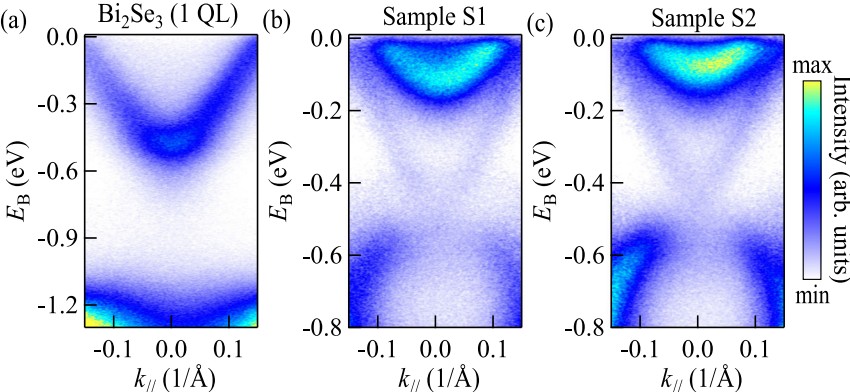

**Fig. 4 | ARPES spectra of the Bi$_2$Se$_3$(1 QL)/X/Si(111) system. a** is the ARPES spectrum of Bi$_2$Se$_3$(1 QL)/Si(111). X in **b** and **c** is Bi$_2$Te$_3$(2 QL) (Sample S1) and Bi$_2$Te$_3$(1 QL)/Bi$_2$Se$_3$(1 QL) (Sample S2), respectively.

third of the experimental one, the EEI dominants in ultrathin films and the influence of the size effect should be treated as a perturbation. Meanwhile, if there are systems where QSH can still be realized even after taking Coulomb repulsion into account, the interaction-induced gaps in such systems should be smaller than the size-effect-induced ones. For example, it may occur in thicker TI films or systems with small $\mu$. On the one hand, we find a better method for fitting the relationship between the mass gap and thickness in Bi$_2$Se$_3$, but this gap is topologically trivial. On the other hand, the ARPES results of samples S1 and S2 hint at being able to treat the surface coupling as a screened Coulomb potential, which makes the analysis of ultrathin films much easier. Our method also makes it possible to analyze the interactions between the high-energy-like particles in condensed matter physics.

## Discussion
Instead of calculating the specific form of surface states by the single-particle energy band theory, we investigate the coupling between two quasi-particle systems using Green function with the screened Coulomb interaction. A set of Bi$_2$Se$_3$ films with different thicknesses are analyzed. The theoretical value of band shifting and the mass gap fit our experimental results well. Our method offers valuable insight into the nature of energy gaps in ultrathin Bi$_2$Se$_3$ films, which may be induced by the breaking of chiral symmetry instead of the time-reversal symmetry. This method also allows us to analyze the Green function in ultrathin TI films much more easily and explains the problems of the sign of the mass gap and the spin polarization near the $\Gamma$ point in previous work perfectly.

## Methods
### Sample preparation and characterization
High-quality Bi$_2$Se$_3$ films were grown on Si(111) substrates by in-situ ultrahigh vacuum (base pressure: $1.0 \times 10^{-10}$ mbar) MBE. Si substrates were firstly annealed to 1000 °C to get Si(111)-7 × 7 reconstructed surface. Then, in order to remove the surface dangling bonds, one monolayer of Bi was deposited on the Si(111)-7 × 7 reconstructed surface and annealed at 300 °C to get the Si(111)-Bi-$\beta$-$\sqrt{3} \times \sqrt{3}$ phase as the wafer layer. High-purity Bi and Se with a ratio of 1:14 were co-evaporated from standard Knudsen cells onto Si(111)-Bi-$\beta$-$\sqrt{3} \times \sqrt{3}$ with the substrate temperature of 220 °C. The Si(111)-7 × 7 reconstructed surface, Si(111)-Bi-$\beta$-$\sqrt{3} \times \sqrt{3}$ surface, and the surface of Bi$_2$Se$_3$ are characterized by low energy electron diffraction pattern. The crystal structure was confirmed by ex-situ X-ray diffraction. The growth rate determined by ex-situ small angle X-ray reflection is ~0.59 QL/min.

### ARPES measurements
To perform ARPES measurements, a Se capping layer with a thickness of 5 nm was deposited to protect the surface from contamination before transferring to another MBE-ARPES system. To probe the

spectral function of Bi$_2$Se$_3$, the samples were heated up to 280 °C and kept for 5 h to remove the Se capping layer and then transferred to the ARPES chamber. The spectral function of the top surface is detected by ARPES with He-$\alpha$I light (21.2 eV) and a Scienta DA30 apparatus as the electron energy analyzer under $T = 5$ K with a base pressure ~$3.0 \times 10^{-11}$ mbar. The pass energy is set to 5 meV and the energy resolution is better than 20 meV, which is confirmed bythe polycrystalline gold standard sample during the ARPES measuring.

## Data availability
The ARPES data generated in this study have been deposited in the figshare database under accession code 23523387. The processed ARPES data are available from the corresponding author upon request. The AFM data generated in this study are provided in the Supplementary Information.

## Code availability
The codes that support this study are available from the corresponding author upon request.

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

## Acknowledgements
This work is supported by the National Key Research Program of China (Grant no. 2022YFA1403302, Z.H.C.), and the National Natural Sciences Foundation of China (Grant nos. 52031015, U22A20115, Z.H.C.). The authors thank Professor Shun-Qing Shen at the University of Hong Kong for his constructive suggestions on the manuscript and Dr. Aeron McConnell at North Carolina State University for his linguistic assistance in the revision of this manuscript.

## Author contributions
Z.H.C. is responsible for the project and conceived this study and the experiments. J.L. performed the theoretical calculations with the assistance of X.Y., W. H., and N.L. The numerical calculation program was written by J.L. and improved by X.Y. and Z.Z.G. J.L., X.Y. and Z.H.C. designed the experiments. J.L., R.S., and H.X. grew the films and performed the basic sample characterization. X.G. made the morphological characterization with AFM. J.L., H.X., and X.Y. performed the ARPES measurements. W.H., X.Q.Z., R.S., Y.L., and Z.K.X fabricated the devices. D.S.X. gave suggestions regarding the theoretical model. J.L., X.Y., and Z.H.C wrote the manuscript. All authors discussed the results and contributed to the manuscript preparation.

## Competing interests
The authors declare no competing interests.
