## [Peer Review File · Nature Communications]

REVIEWER COMMENTS

Reviewer #1 (Remarks to the Author):

The authors report theoretical and ARPES studies of surface coupling in Bi₂Se₃ thin films. The thickness dependence of a set of Bi₂Se₃ thin films is examined by ARPES measurement and shows the mass gap induced by the surface coupling. These experimental results also fit well with theory, and demonstrate an accurate means to describe thin film systems. The results are well presented, and the paper is well written. However, a few questions need to be addressed and some of the supplementary data is missing.

1. In Fig. 2, the authors say that the energy gap opened below 6 QL. However, aside from the light green dots and red dot line, the raw ARPES data is not clear enough to identify the gap opening. For example, even for 5 QL, the gap actually seems to be still closed. The data quality is also not as clear as the cited work (reference 13).
2. In Fig. 2, the light green dots and red dot lines are all fitting results, and the main claim of this manuscript relies on these data. The authors should provide more details explaining the fitting procedure to get these results in the supplementary materials.
3. The authors compare the experimental results with theory, which involves lots of calculations and numerical results, and these are briefly discussed in the main text. The authors should consider adding more details about the calculations in the supplementary materials to support the theoretical results in the main text.
4. A minor note that, in line 38, "can open" is repeated.

Reviewer #2 (Remarks to the Author):

This paper addresses, both theoretically and experimentally (ARPES) a very timely issue, namely the role of many-body effects on the spectra of purported topological insulators.

In several previous works, the effects on the mass gap in the single-particle spectrum of such systems has been studied as a function of thickness of samples, i.e as a function of quintuplet layers (QL). It has been found that this mass gap can change sign from positive (topologically trivial) to negative (topologically non-trivial) as thickness is varied. The authors of the present paper point out that i) the gap reclosing (necessary for a change of sign in m) has not been observed, ii) the experimentally reported mass gaps are far larger than predicted, and iii) the absence of ballistic 1D edge states means that the system is topologically trivial.

In order to understand the situation better, the authors compute, using a self-consistent procedure, the mass taking inter-surface (top and bottom) screened Coulomb-interactions into account. They find within their self-consistent scheme that the mass does not change with thickness, although it does get very small with increasing thickness. They then go on to present ARPES data on the spectra of the Bi₂Se₃-system they consider, and conclude that a visible gap opens up for thicknesses around 6 QL and that the system is topologically trivial.

I have several comments to this paper.

1. The mass they compute does not change sign but becomes extremely small rapidly with increasing thickness. For instance, by the time one gets to 11 QL, it would be down by a factor of roughly $e^{-8.5} = 0.0002$. In ARPES, one does not get the sign of m , only the absolute value. ARPES is not suited for looking for sign changes in m , and can therefore not rule out sign changes either.
2. Their Eq. 10 is fitted to data in Fig. 3. Given the smallness of the mass, how confident should one be in the estimate, and the sign, given the many approximations involved both in the way they do their self-consistent calculation, as well as to factors ignored by the self-consistent approach?
3. It is rare to see papers in such high-profile journals as Nature written in such poor English. The manuscript is rife with basic grammatical errors and poorly formulated sentences throughout. The authors need to remedy this before resubmitting it to any any scientific journal using the English

language.

As this paper stands, I do not recommend it for publication in Nature Communications.

RESPONSE TO REVIEWER COMMENTS

Response to Reviewer #1:

Overall comments:

'The authors report theoretical and ARPES studies of surface coupling in Bi₂Se₃ thin films. The thickness dependence of a set of Bi₂Se₃ thin films is examined by ARPES measurement and shows the mass gap induced by the surface coupling. These experimental results also fit well with theory, and demonstrate an accurate means to describe thin film systems. The results are well presented, and the paper is well written. However, a few questions need to be addressed and some of the supplementary data is missing.'

Authors' response: We greatly thank the reviewer for her/his positive overall comments on our work. Below we addressed all the concerns raised by the reviewer. We also add the detailed fitting procedures and new data in the revised supplementary information.

Comment 1.1:

'In Fig. 2, the authors say that the energy gap opened below 6 QL. However, aside from the light green dots and red dot line, the raw ARPES data is not clear enough to identify the gap opening. For example, even for 5 QL, the gap actually seems to be still closed. The data quality is also not as clear as the cited work (reference 13).'

Authors' response: We have tried our best to obtain high-quality ARPES data. During the ARPES measurements, we used a quite small pass energy (~5 meV) to improve energy resolution, which reduces the collection efficiency of photoelectrons. To improve the spectra visually, we change the color bar of ARPES spectra. We further resize the final ARPES spectra and calculate the curvature distribution near 5 QL (Fig. R1a-c) [Ref. *Rev. Sci. Instrum.* **82**, 043712 (2011)], which are shown in Fig. S2a-c in the revised supplemental information. To get the magnitude of mass gaps more precisely, the energy distribution curves (EDCs) at Γ points are fitted by Lorentzian peaks as shown in Fig. S3 (Fig. R2). Although the energy gap is quite small near 5 QL, it can be evidently identified. Besides the fitting error of the peak position, we also added the full width at half-maximum (FWHM) of the fitting results as the additional error in the error bars in Fig. 3. The mass gaps extracted by the previous work you mentioned [*Nat Phys* **6**, 584-588 (2010)] are also added in Fig. 3b (Fig. R3) for comparison.

Comment 1.2:

'In Fig. 2, the light green dots and red dot lines are all fitting results, and the main claim of this manuscript relies on these data. The authors should provide more details explaining the fitting procedure to get these results in the supplementary materials.'

Authors' response: We have provided more details explaining the fitting procedure to get these results in the revised supplementary materials. Moreover, the momentum distribution curves (MDCs) and energy distribution curves (EDCs) of ARPES spectra with the fitted Lorentzian peak positions are also shown in Fig. R1 (Fig. S2), which are the black dots come from in Fig. 1c and Fig. 2 in main text. The red dot lines are the fitting result of these black dots by the Dirac dispersion relation $E_{\pm}(k) = \mu \pm \sqrt{(\hbar v_F k)^2 + m^2}$.

Comment 1.3:

'The authors compare the experimental results with theory, which involves lots of calculations and numerical results, and these are briefly discussed in the main text. The authors should consider adding more details about the calculations in the supplementary materials to support the theoretical results in the main text.'

Authors' response: We have added more details about the calculations in the revised supplementary materials to support the theoretical results in the main text. The error bar induced by each approximation is also discussed.

Comment 1.4:

"A minor note that, in line 38, 'can open' is repeated."

Authors' response: We have checked all grammar errors and misspellings in the revised manuscript and supplementary materials. We asked one colleague whose native language is English to improve the language in the revised manuscript.

Fig. R1. The peak positions fitted from ARPES spectra of different thicknesses Bi_2Se_3 . **a-c** are the improved ARPES spectra of 4-6 QL Bi_2Se_3 by the method of curvature distribution [Ref. Rev. Sci. Instrum. 82, 043712 (2011)]. **d-j** and **k-o** are MDCs and EDCs of smoothed ARPES spectra of Bi_2Se_3 whose thickness is labeled at the right-top of each subfigure. The red short lines are the peak positions fitted by Lorentzian peak.

Fig. R2. The fitting results of EDCs at Γ point. a-d are the fitting results in the thickness of 2-5 QL. The black lines are the EDCs of the ARPES spectra, while the orange lines and red lines are the Lorentzian fitting of the single peak and the total EDCs. The FWHM of fitting results is also considered as an additional error in the error bar in Fig. 3.

Fig. R3. The experimental and theoretical results of Dirac position μ and the magnitude of gap $2m$, respectively. The cyan circles in b are the results of Y. Zhang, et al.

Response to Reviewer #2:

Overall comments:

'This paper addresses, both theoretically and experimentally (ARPES) a very timely issue, namely the role of many-body effects on the spectra of purported topological insulators.

'In several previous works, the effects on the mass gap in the single-particle spectrum of such systems has been studied as a function of thickness of samples, i.e., as a function of quintuplet layers (QL). It has been found that this mass gap can change sign from positive (topologically trivial) to negative (topologically non-trivial) as thickness is varied. The authors of the present paper point out that i) the gap reclosing (necessary for a change of sign in m) has not been observed, ii) the experimentally reported mass gaps are far larger than predicted, and iii) the absence of ballistic 1D edge states means that the system is topologically trivial.

'In order to understand the situation better, the authors compute, using a self-consistent procedure, the mass taking inter-surface (top and bottom) screened Coulomb-interactions into account. They find within their self-consistent scheme that the mass does not change with thickness, although it does get very small with increasing thickness. They then go on to present ARPES data on the spectra of the Bi_2Se_3 -system they consider, and conclude that a visible gap opens up for thicknesses around 6 QL and that the system is topologically trivial.

'I have several comments to this paper.'

Authors' response: We greatly thank the reviewer for her/his positive overall comments on our work. We made some improvement in revised manuscript. The supplemental information with the details of theoretical derivation, analysis of error and the energy band fitting is also added. Below we addressed all the concerns raised by the reviewer.

Comment 2.1:

'The mass they compute does not change sign but becomes extremely small rapidly with increasing thickness. For instance, by the time one gets to 11 QL, it would be down by a factor of roughly $e^{-8.5} = 0.0002$. In ARPES, one does not get the sign of m , only the absolute value. ARPES is not suited for looking for sign changes in m , and can therefore not rule out sign changes either.'

Authors' response: m does not change its sign with our theoretical and experimental results. It is true that ARPES is not a suitable method to confirm the sign of m in the Dirac system since the energy band $E_{\pm}(k) = \mu \pm \sqrt{(\hbar v_F k)^2 + m^2}$ does depends on the absolute value, rather than the sign of m . We rule out the sign change based on the following calculation and experimental results:

1. The error analysis (see our response of the next comment) confirms that the sign of m will not change in the case of our mass theory.

2. If the sign of m changes, the absolute value of m will evolve non-monotonically, which can be detected by our ARPES. For example, the $(\text{Bi}_{1-x}\text{In}_x)_2\text{Se}_3$ with thickness of larger than 6 QL show a non-monotonic evolution with In-doping as shown in Fig. R4a [Nano Lett. 19, 4627 (2019)]. However, this non-monotonic evolution does not appear in our ARPES results and previous work [Nat. Phys. 6, 584-588 (2010)]. To further confirm this result, we extracted the energy gap in the Bi_2Se_3 films with the thickness of 1.5 and 2.5 QL in reviewed manuscript, and the phenomenon of gap reclosing at 2.5 QL predicted by previous theory is not observed as shown in Fig. R4b (Fig. 3b).
3. The changing sign of m corresponds to a transform between the topologically trivial phase and the Z_2 topological phase [Phys. Rev. B 81, 115407 (2010). Phys. Rev. B 81, 041307(R) (2010)]. However, the previous work [Nano Lett. 19, 4627 (2019).] has shown the trivial topological nature of Bi_2Se_3 below 6 QL (Fig. R4c-d). The 1D ballistic transport channel has also been confirmed to be absent in such systems [Adv. Quantum Technol., 2200043 (2022)]. A non-zero m (no need to be negative) will cause a spin polarization σ_z in the z-direction near Γ point [Phys. Rev. B 81, 115407 (2010)], which was not observed in the spin-resolved ARPES measurement [Nat. Phys. 8 616 (2012)].

All listed experimental results demonstrate that the single-particle band theory is not sufficient to describe the band structure and transport properties in ultrathin Bi_2Se_3 TI films. Therefore, we suspect that the mass gap may come from the influence of the Coulomb interaction. Within the framework of our theory, the energy gap originates from the breaking of chiral symmetry instead of the time-inversion symmetry for one band of a Kramer pair, which is similar to the case in graphene with electron-electron interaction and the Kekulé-Ordered graphene [Phys. Rev. Lett. 126, 206804 (2021)].

For the 5 QL Bi_2Se_3 film, the measured mass gap is about 40 meV. The mass gaps for the samples with thickness above 6 QL are much smaller. Such small gap can be easily disturbed by the scattering and temperature. Even in the case of quantum anomalous Hall effect [Science 329, 659 (2010)., Science 340, 167 (2013).], the magnetic-doping induced gap is about 50 meV, which needs the low temperature of 30 mK for the transport measurements. Therefore, the magnitude of the energy gaps for the samples with thickness of above 6 QL cannot be detected, and will be not discussed here.

Comment 2.2:

‘Their Eq. 10 is fitted to data in Fig. 3. Given the smallness of the mass, how confident should one be in the estimate, and the sign, given the many approximations involved both in the way they do their self-consistent calculation, as well as to factors ignored by the self-consistent approach?’

Authors' response: We really thank the referee for this illuminating comment, which pushed us to find a new and more accurate approximation in the revised manuscript. All used approximations and the details of calculations are shown in the revised supplemental information. The relative errors of each approximation are also discussed, and the largest one is less than 3.5% (about 10 meV at the thickness of 2QL and even smaller at others, which is smaller than the energy resolution of our ARPES). All approximations we made will only affect the magnitude of the final results, and the sign of m does not change. Here we list all methods of approximations with relative error.

1. We ignored the wave-function renormalization functions $A_{1,2}$ in Eq. (S1.3), and the mass term in Eq. (S2.5) is treated as a real number $m(ik_n, k) \approx m(0,0) = \text{Const}$. This approximation reduced the accuracy of the self-energy when $k \neq 0$, but has much less impact on the problem of the interaction-induced mass gap at the Gamma point and is widely used [references 1-5 in supplemental information].
2. The Coulomb interaction in our system is hard to calculate, and we find an expression as Eq. (S1.1) which deviates very little (relative error $<10^{-5}$) from the numerical result (see Fig. S1).
3. Zero-temperature approximation is used to simplify the polarization function in Eq. (S2.13). Our experiments were conducted at temperature of 5 K, which correspond to a 4 meV broadening of Fermi-Dirac distribution from 1% to 99% at the Fermi level. This broad is much smaller than all energy scales that we discussed, and therefore can be ignored. We also notice that the ARPES spectra were measured at room temperature in previous work [*Nat. Phys.* **6**, 584-588 (2010)] and their fitting results of the mass gap are quite close to ours, which implies that the mass gap will not sensitively change with temperature.
4. The Lorentzian peak $\frac{1}{\pi} \frac{\eta}{x^2 + \eta^2}$ is approximated as a delta-function $\delta(x)$ in the integral of \mathbf{q} in Eq. (S1.4). Compared with previous manuscript, this approximation has been improved. To ensure the accuracy of this approximation, the full-width at half maximum (FWHM) of this Lorentzian peak η should as little as possible. In previous manuscript, we treat the $\frac{q}{q^2 + \lambda^{-2}}$ term in Eq. (S1.3) as

$$\frac{q}{q^2 + \lambda^{-2}} = \frac{\pi}{\sqrt{2\lambda^{-1}}} \frac{1}{\pi \left(\sqrt{q} - \frac{\lambda^{-1}}{\sqrt{q}} \right)^2 + 2\lambda^{-1}} \approx \frac{\pi}{2\lambda} \delta(q - \lambda^{-1}),$$

with FWHM= $\sqrt{2\lambda^{-1}}$. We use the new method in the revised manuscript to treat the term $\frac{q}{q^2+\lambda^{-2}}$ as

$$\frac{q}{q^2 + \lambda^{-2}} = \frac{\pi q^2 + \lambda^{-2}}{2} \frac{\lambda^{-1}}{\lambda^{-1}} \frac{\partial}{\partial q} \left[\frac{1}{\pi} \frac{\lambda^{-1}}{q^2 + \lambda^{-2}} \right] \approx \frac{\pi q^2 + \lambda^{-2}}{2} \frac{\lambda^{-1}}{\lambda^{-1}} \delta(q)$$

with FWHM= $\lambda^{-1} \approx 0.03 \text{ \AA}^{-1}$. The latter one has a narrower linewidth. The fluctuation of $f(q)$ near $q = 0$ in Eq. (S1.4) originates from the Green function

$$\frac{m}{(-iq - \mu_0)^2 + (m^2 + \hbar^2 v_F^2 q^2)} \simeq \frac{m}{(-iq - \mu_0)^2 + m^2} \left[1 - \frac{\hbar^2 v_F^2 q^2}{(-iq - \mu_0)^2 + m^2} \right]$$

Therefore, the relative error of delta-function approximation is less than $\frac{\hbar^2 v_F^2 \lambda^{-2}}{\mu_0^2} \approx 3.5\%$.

Meanwhile, in the approximation methods above, all the omitted parts do not contain the opposite sign, which makes the final result will not change the sign of m .

Comment 2.3:

‘It is rare to see papers in such high-profile journals as Nature written in such poor English. The manuscript is rife with basic grammatical errors and poorly formulated sentences throughout. The authors need to remedy this before resubmitting it to any scientific journal using the English language.’

Authors’ response: We have checked all grammar error and misspellings in the revised manuscript and supplementary materials. We asked one colleague whose native language is English to improve the language in the revised manuscript.

Fig. R4. **a** The energy gap evolution of Bi₂Se₃ with different thicknesses vs. the In-doping [*Nano Lett.* **19**, 4627 (2019)]. **b** Our result of the energy gap evolution of Bi₂Se₃ vs. thickness. **c, d** are the z-component of spin polarization in 3 QL Bi₂Se₃ measured by spin-ARPES and the corresponding schematic of spin texture [*Nat. Phys.* **8** 616 (2012)].

LIST OF CHANGES

All changes are highlighted in the manuscript, and we list them here one by one.

1. The supplemental information with the details of the derivation of our theory and the peak fitting of ARPES spectra is added in the additional file.
2. Line 1 (In the reviewed manuscript, same below): ‘Bi₂Se₃ thin films’ is changed to ‘Bi₂Se₃ ultrathin films’ in the title.
3. Line 14-15: ‘the transport properties and band structure in ultrathin TI films because of the upper- and lower-surface states coupling.’ is changed to ‘their band structures and transport properties due to the existence of top and bottom surface states coupling.’
4. Line 15: ‘the upper- and lower-surface states coupling’ is changed to ‘this coupling’.
5. Line 18-20: ‘We find that the mass gap induced by the surface state coupling is always positive, which makes the surface state topological trivial, and the magnitude of mass gap is in good agreement with the experimental results.’ is changed to ‘We find that in the case of thickness below 6 quintuple layers (QLs), the magnitude of the mass gap is in good agreement with the experimental results.’
6. Line 24-25, 31, 93, 149-150: ‘because of’ is changed to ‘owing to’.
7. Line 26-27: ‘with different dispersion relations and corresponding spin-momentum locking’ is changed to ‘, such as Dirac fermions, type-I and type-II Weyl fermions, as well as corresponding spin-momentum locking². Many novel properties induced by the surface states have been investigated extensively’.
8. Line 30: ‘In most cases, these topological surface states are housed within a few nanometers of the surface of samples’ is changed to ‘These topological surface states are generally housed within a few nanometers of the surfaces of samples’.
9. Line 32-33: ‘Therefore, a large surface-volume ratio is necessary to amplify the dominance of surface state in the whole sample, which is achievable in ultra-thin films.’ is changed to ‘Ultrathin TI films with a large surface to volume ratio are ideal systems to amplify the dominance of surface states.’
10. Line 33-34, 59, 78, 82, 86, 139, 177, 186: ‘upper- and lower-surface states’ is changed to ‘top and bottom surface states’.

11. Line 35: ' $E_{\pm}(\mathbf{k}) = -E_D \pm \sqrt{(\hbar v_F)^2 k^2 + m^2}$ ' is changed to ' $E_{\pm}(\mathbf{k}) = \mu \pm \sqrt{(\hbar v_F)^2 k^2 + m^2}$ '
12. Line 37: 'For the thicker Bi₂Se₃ samples' is changed to 'For the Bi₂Se₃ samples with thickness above 6 quintuple layers (QLs)'.
13. Line 38-39: 'while the surface states coupling can open a gap can open in ultra-thin films.' is changed to 'while the coupling between top and bottom surface states can open an energy gap for the samples with thickness of below 6 QL, i.e. $m \neq 0$ '.
14. Line 39-43: 'It should be noticed that the Chern number, ..., while a negative m represents a nonzero Hall conductance with a gapless helical edge state^{1, 11}.' is changed to 'It should be noted that, ..., depends on the sign of mass gap m ^{1, 11}'.
15. Line 44-58: We made another paragraph and rewrite the second paragraph from the previous manuscript from

'In previous works on Bi chalcogenide ultra-thin films^{11, 12}, ... Inspired by the fact that the electron-electron interaction (EEI) will break the chiral symmetry and open a mass gap in graphene^{18, 19}.'

to

'In previous works on Bi₂Se₃ ultrathin films^{11, 12}, ... It seems that the single-particle band theory is not sufficient to describe the band structure and transport properties in ultrathin Bi₂Se₃ TI films.'
16. Line 61-62: 'The band shifting and the mass gap of Bi₂Se₃ films with different thicknesses are also extracted from ARPES spectra. The theoretical and experimental results fit well, and the problem of sign of mass gap is also solved.' is changed to 'The band shifting and the mass gap of Bi₂Se₃ films with different thicknesses are extracted from ARPES spectra and fit the theoretical results well.'
17. Line 62-67: We added 'Within the framework of our theory, ..., which is consistent with reported experimental results.'
18. Line 71-73: 'The Dirac fermion is described by Dirac Hamiltonian $H = v\alpha \cdot \mathbf{k} + \beta m - E_D$, ..., v is the Fermi velocity and m is the mass gap at $\mathbf{k} = 0$ ²⁰.' is changed to 'The Hamiltonian of massless Dirac fermions is $H = \hbar v_F \alpha \cdot \mathbf{k} - \mu_0$, ..., and \hbar is the reduced Plank constant.'
19. Line 74: 'the upper and lower surface states' is changed to 'the top ($\alpha = +1$) and bottom ($\alpha = -1$) surface states ψ_{α} '. We deleted 'where $\alpha = \pm 1$ represent the upper (+) and lower

- (-) surface state,' and 'Dirac'.
20. Line 79: ', and the' is deleted.
 21. Line 81: 'The Fermi velocity of Dirac fermion in Dirac material v_F is about $\sim 10^5$ m/s' is changed to 'The Fermi velocity, v_F , of Dirac fermions in Dirac material is about $\sim 10^5$ m/s'
 22. Line 84: $\frac{e^2}{2\epsilon_0}$ is changed to $\frac{e^2}{4\pi\epsilon_0}$.
 23. Line 86-87: 'The 2-dimentional Fourier transformation of $V(r, d)$ with respect to \mathbf{r} is difficult to calculate.' is changed to 'Since it is difficult to calculate the 2-dimensional Fourier transformation of $V(r, d)$ with respect to \mathbf{r} ,
 24. Line 87: 'find' is changed to 'adopt'.
 25. Line 90-92: 'whose relative error is less than 10^{-5} compared with the numerical results.' is changed to 'To analyze the error induced by this approximation, ... The relative error $\left| \frac{V_{\text{num}} - V}{V_{\text{num}}} \right|$ is less than 10^{-5} (see supplemental information S.I).'
 26. Line 93: 'in real samples' is changed to 'present in the fabricated samples'.
 27. Line 94: v_F is changed to $\hbar v_F$ and μ_0 is changed to μ .
 28. Line 96: 'and $A_{1,2}$ denotes' is changed to ' $A_{1,2}$ was denoted as'.
 29. Line 96-101: We added ' $ik_n = i(2n + 1)\pi/\beta$ is the Matsubara frequency, ..., we focus on the calculation of the first two quantities μ and m .'
 30. Line 102-103: 'in the sample much thicker than the screening length λ experimentally, such as the ARPES spectra' is changed to 'experimentally by ARPES spectra in a thick enough sample'.
 31. Line 103-105: We switched the order of 'which is approximately equal to half of the critical thickness when the energy gap opens²³' and 'can be calculated by the density functional theory numerically.'
 32. Line 109: $V(0, d)$ is changed to $V(q = 0, d)$.
 33. Line 111-115: We added 'Since we are only concerned with the mass gap at the Γ point, ..., and consequently obtain the self-consistent gap equation as follows'
 34. Line 116: The step of taking the trace is added as $\frac{1}{N_\gamma} \text{Tr}$.
 35. Line 117: We added 'where N_γ is the rank of chosen gamma matrices'.
 36. Line 117-119: We switched the order of ' $\Pi_{-\sigma}(iq_m, \mathbf{q})$ is the polarization function of the Dirac fermion in the $-\sigma$ surface defined as' and the Eq. (7). The symmetric factor -2 of

polarization is also fixed.

37. Line 120: ‘schematic’ and ‘Since only the sign and the magnitude of the mass gap at Γ point are mainly concerned, it is reasonable to simplify the 2+1 dimensional mass gap function as a constant $m(ik_n, \mathbf{k}) \approx m$ ’ are deleted, and ‘in this condition’ is changed to ‘of Dirac fermion’.
38. Line 121-122: ‘studied’ is changed to ‘investigated’, and ‘and the similar form of polarization function in our condition reads’ is changed to ‘We calculate the polarization function as follows (see supplemental information S.II)’
39. Line 123: \hbar^2 is added.
40. Line 123: The off-diagonal part is added, and the $C = -iq_m\mu_0(1 - 2x) + 2m^2$ is changed to $C = -iq_m\mu_0(1 - 2x) + 2m^2 + m\hbar v_F \boldsymbol{\gamma} \cdot \mathbf{q} + m[(1 - 2x)iq_m - 2\mu_0]\gamma_0 + [2x(1 - x)iq_m + (1 - 2x)\mu_0]\hbar v_F \boldsymbol{\gamma} \cdot \mathbf{q}\gamma_0$.
41. Line 126: $\mu_0 < B < \mu_0$ is changed to $-\mu_0 < B < \mu_0$.
42. Line 130: ‘ $\frac{e^2}{16\sqrt{2}\pi\epsilon_0^2} e^{-2\sqrt{2}d/\lambda} g(q = \lambda^{-1})$ ’ is changed to ‘ $\frac{e^4}{64\pi^2\epsilon_0^2} \left[\frac{1}{\pi} g(q = 0) + \lambda^{-1} \frac{\partial}{\partial q} \Big|_{q=0} g(q) \right] e^{-2\lambda^{-1}d}$ (see supplemental information S.I)’.
43. Line 132-133: ‘Finally, one can find the thickness-dependent mass gap’ is changed to ‘Finally, one can obtain the thickness-dependent mass gap (See the supplemental information S.II regarding the detailed error analysis) as:’
44. Line 134: $m \approx \frac{e^4\mu}{128\sqrt{2}\pi^2\epsilon_0^2 v_F^2} \exp\left[-\frac{2\sqrt{2}d}{\lambda}\right]$ is changed to $m \approx \frac{e^4\mu}{128\pi^3\epsilon_0^2 \hbar^2 v_F^2} \exp\left[-\frac{2d}{\lambda}\right]$.
45. Line 136: ‘which makes the problem of gap reclosing fixed’ is changed to ‘and consequently it is consistent with the absence of gap reclosing probed from ARPES.’
46. Line 140: ‘quintuple layers (QLs)’ is changed to ‘QL’.
47. Line 141: ‘are well described’ is changed to ‘were described’.
48. Line 142-143: ‘decay with the thickness with the decay length $\lambda \approx 3$ QL’ is changed to ‘decays exponentially with the thickness at the rate of the surface state decay length $\lambda \approx 3$ QL’.
49. Line 143: We added ‘consequently’.
50. Line 144: v is changed to v_F .
51. Line 145: ‘Fig. 1b-c by following steps’ is changed to ‘Fig. 1b using the following steps’.

52. Line 146-148: ‘as shown in Fig. 1c’ is changed to ‘(see supplemental information S.III)’. ‘and the peak position is exactly located where the energy band pass through. ... The fitting results is’ is changed to ‘and the fitted peak positions are marked as black dots in Fig. 1b. Then we linearly fit these two sets of peak positions as shown by red dotted lines in Fig. 1b and obtain’.
53. Line 149: We deleted ‘As shown in Fig. 1b,’ and v is changed to ‘ $\hbar v_F$ ’.
54. Line 151-152: ‘which is similar to previous work’ is changed to ‘These values are in good agreement with those of the previous work’.
55. Line 152-153: ‘the dispersion relation of the generalized Green function $E_{\pm}(k) = \mu \pm \sqrt{\hbar^2 v^2 + m^2}$ is used because of the existence of energy gap $E_{\text{gap}} = 2m$.’ is changed to ‘we used the dispersion relation of the generalized Green function $E_{\pm}(\mathbf{k}) = \mu \pm \sqrt{\hbar^2 v_F^2 k^2 + m^2}$ due to the existence of energy gap $E_{\text{gap}} = 2m$.’
56. Line 153-155: ‘We check whether there is an energy gap by the energy distribution curve (EDC) at the Γ point, and determine the magnitude of m by fitting the peak positions of this curve.’ is changed to ‘By fitting the peak positions we can check whether or not there is an energy gap in the energy distribution curve (EDC) at the Γ point, and further determine the magnitude of m .’
57. Line 156: We added ‘More details on peak fitting are present in the supplemental information S.III.’ and broke the following clauses into a separate sentence.
58. Line 158: We change ‘scatter’ to ‘scatter plots’ and added ‘, respectively’.
59. Line 158-161: ‘The energy gap opened below 6 QL consistent with previous work¹³, and the phenomenon of energy gap reclosing described in the previous theory is not found¹¹.’ is changed to ‘To confirm whether the gap will reclose or not at about 2.5 QL predicted by the previous theory¹¹, ..., the non-monotonic evolution of the energy gap around the thickness of 2-3 QL is not found.’
60. Line 162: ‘Obviously,’ is changed to ‘We can clearly see that’.
61. Line 163-165: ‘However, it has been shown that the band bending effect at the interface between Bi_2Se_3 and vacuum is less than 0.1 eV when the thickness is less than 7 QL²⁵. The band bending effect is much smaller at the interface between Bi_2Se_3 and vacuum.’ is changed to ‘However, ..., and it should be much smaller at the interface between Bi_2Se_3 and silicon.’
62. Line 165-167: ‘the band shifting cannot be explained by the band bending effect because it is larger than 0.1 eV as long as the thickness is thinner than 6 QL.’ is changed to ‘As shown in

Fig. 3a, ..., therefore cannot be explained by the band bending effect.'

63. Line 167: We deleted 'by red line'.
64. Line 169: We deleted 'by red line'.
65. Line 169-176: We rewrote the least part of this paragraph from 'in which one can find that the curve also fits experimental result well and fix the problem of energy gap reclosing..., which is smaller or similar than the energy resolution of most ARPES and the band broadening in most samples²⁶.' to 'We can find that the curve fits both our experimental results and previous work as the cyan circles in Fig. 3b for comparison¹³..., which is smaller than the energy resolution of ARPES or the band broadening in most cases²⁹.'
66. Line 176: We deleted 'With the method above, there is'.
67. Line 178: 'come in' is changed to 'are based on'.
68. Line 179: 'fact' is changed to 'method'.
69. Line 182: The reference of λ_{BT} is added.
70. Line 183: We added 'the value found in'.
71. Line 188-192: 'In fact, ..., we calculated the self-consisted mass gap equation.' is changed to 'Interestingly, ..., the EEI dominants in ultrathin films and the influence by the size effect should be treated as a perturbation.'
72. Line 192: 'fitting of' is changed to 'method for fit', and 'relation' is changed to 'relationship'.
73. Line 194: 'and the theoretical results show the fact that this gap is topologically trivial' is changed to 'but this gap is topologically trivial'.
74. Line 196: 'This' is changed to 'Our'.
75. Line 200: 'by the method of' is changed to 'using'.
76. Line 202-207: We rewrote this part from 'The band shifting and the mass gap is calculated by the Hartree self-energy and self-consisted gap equation with three parameters determined experimentally or numerally. ... This method makes it much easier to analyze the Green function in ultra-thin TI films and perfectly explain the previous problems of the sign of mass gap and the open boundary condition in previous work.' to 'A set of Bi_2Se_3 films with different thicknesses are analyzed. ... This method also allows us to analyze the Green function in ultrathin TI films much more easily and explains the problems of the sign of the mass gap and the spin polarization near the Γ point in previous work perfectly.'

77. Line 210: ‘As the first step, the Si substrates were annealed to 1000 °C’ is changed to ‘Si substrates were firstly annealed to 1000 °C’.
78. Line 211: ‘to’ is changed to ‘in order to’.
79. Line 220: We deleted ‘state’ and changed ‘MBE’ to ‘MBE-ARPES system’.
80. Line 222: We changed ‘capper’ to ‘capping layer’ and changed ‘up-’ changed to ‘the top’.
81. Line 224: We added ‘The pass energy is set to 5 meV and’.
82. Line 231-232: We added the code availability statement.
83. Line 234-239: We rewrote the acknowledgements from ‘This work is supported by the National Key Research Program of China, ... The authors thank Prof. Shun-Qing Shen at the University of Hong Kong for his constructive suggestions on the manuscript.’ is changed to ‘This work is supported by the National Key Research Program of China ... Dr. Aeron McConnell at North Carolina State University for his linguistic assistance in the revision of this manuscript.’
84. Line 292-296: We added ‘Wang, Z., *et al.* ... *Nano Lett* **19**, 4627-4633 (2019).’ and ‘Xu, S.-Y., *et al.* ... *Nat Phys* **8**, 616-622 (2012).’ as the 14th and 15th reference, respectively. The serial number of subsequent references should be added by 2.
85. Line 316-317: We changed ‘Lv, BQ, ... *Rev. Mod. Phys.* **93**, (2021).’ to ‘Appelquist, T. W. ... *Phys Rev D* **33**, 12 (1986).’
86. Line 322-326: We switched the order of ‘Li. W., ... *Phys Lett A* **374**, 2957 (2010)’ and ‘Gorbar, E. V., ... *Phys Rev B* **66**, 045108 (2002).’
87. Line 328-329: We added ‘Pisarski, R. D. ... *Phys Rev D* **29**, 10 (1984).’ The serial number of subsequent references should be added by 1(+2).
88. Line 340-341: We switched reference ‘Zhou, B, ... *Phys. Rev. Lett.* **101**, 246807 (2008)’ to ‘Bao, C., *et al.* ... *Phys Rev Lett* **126**, 206804 (2021).’
89. Line 343: We changed the color bar of Fig. 1b., and put the Fig. 1c into the supplemental information. We fix the mistake that Fig. 1b was the ARPES spectral of 7 QL instead of 27 QL.
90. Line 347-349: ‘**b** and **c** ARPES spectra and MDCs of Bi₂Se₃ sample of 27 QL. ..., and the red dot line is the linear fitting results of two sets of light green dots with the same intercept and slope with opposite sign.’ is changed to ‘**b** ARPES spectra of 27 QL Bi₂Se₃ sample. ..., and the red dotted line is the linear fitting results of two sets of black dots.’
91. Line 351-356: We changed the color bar of Fig. 2, and the ‘light green dots’ is changed to the

‘black dots’.

92. Line 358-362: We increased the thickness of the red lines in the two subgraphs for a better impression in Fig. 3. The red line in Fig. 3b is recalculated by the fixed Eq. (10). ‘The cyan circles in b are the results of Y. Zhang, *et al*¹³.’ is added.
93. Line 363-366: We changed the color bar of Fig. 4.
94. Some typo and grammar mistakes are fixed as shown by yellow highlight in the following positions: Line 17, 26, 59, 60, 61, 81, 92, 95, 101, 110, 111, 121, 127, 128, 131, 135, 143, 145, 146, 163, 167, 180, 182, 185, 192, 195, 199, 209, 212, 215, 346, 361.

REVIEWER COMMENTS

Reviewer #1 (Remarks to the Author):

The authors have addressed my questions. I would recommend the publication.

Reviewer #2 (Remarks to the Author):

The authors have adequately addressed the issues that I raised in my first report. I appreciate the effort they have made to improve their analysis of extracting the gap from their data, as well as their efforts in improving the style of the presentation.

I therefore have no objections to the paper being published in Nature Communications.

Reviewer #3 (Remarks to the Author):

The authors have addressed most of the criticisms by the previous reviewers. But, related to the reviewer #1's comment 1.1, it would be good to include additional data and discussion on the following point.

The weak intensity of the surface states in Fig. 4b and 4c makes it unclear whether the present theory can quantitatively explain the difference in the mass of the surface states in multilayered systems. Although the authors say "This method makes the analysis of coupling between surface states intuitive and concise, especially in multilayered systems.", the applicability of their theory to multilayered systems is questionable.

In addition, I have few comments to the manuscript.

1. What can the authors learn from this theory for the surface band structure of 1 QL Bi₂Se₃? For example, how large is the predicted mass at 1 QL, and whether the electron pocket in Fig. 4a is attributed to the upper branch of the surface state? Probably, the band structure of 1 QL films was not well understood in previous studies. New information would be appreciated.

2. If the authors utilize the data of 2.5 QL as a basis to exclude a phase transition between 2 QL and 3 QL, a more detailed explanation on it (and 1.5 QL) would be necessary. What state is 2.5 QL (or 1.5 QL film)? A mix of 2 QL and 3 QL islands? In relation, why are there more peaks in 2.5 QL film in Fig. S3 than in 2 QL or 3 QL? The band dispersion at 2.5 QL (and 1.5 QL) should be plotted somewhere. In addition, not only the mass but also the value of μ for 2.5 QL and 1.5 QL should be compared to the calculated one in Fig. 3.

3. Within this theory, is it generally difficult to achieve the QSH phase in ultra-thin films of any TI? If there are systems where QSH can still be realized even after taking into account Coulomb repulsion, can the authors predict what parameter region they are in?

In conclusion, the subject of this manuscript is important and would attract interests of the researchers in topological material sciences, but the points described above need to be addressed by the authors before I recommend publication.

RESPONSE TO REVIEWER COMMENTS

Response to Reviewer #3:

Overall comments:

'The authors have addressed most of the criticisms by the previous reviewers. But, related to the reviewer #1's comment 1.1, it would be good to include additional data and discussion on the following point.

'The weak intensity of the surface states in Fig. 4b and 4c makes it unclear whether the present theory can quantitatively explain the difference in the mass of the surface states in multilayered systems. Although the authors say "This method makes the analysis of coupling between surface states intuitive and concise, especially in multilayered systems.", the applicability of their theory to multilayered systems is questionable.

'In addition, I have few comments to the manuscript.'

Authors' response:

Thank you for your careful reading and your comments. As shown in Fig. R1b and 1c (Fig. 4b and 4c in main text), we have improved the quality of ARPES spectra of multilayered systems in the revised manuscript. The intensity of the surface states in the revised Fig. 4b and 4c is strong enough to explain the difference in the mass of the surface states in multilayered systems on the basis of present theory.

Comment 3.1:

'What can the authors learn from this theory for the surface band structure of 1 QL Bi₂Se₃? For example, how large is the predicted mass at 1 QL, and whether the electron pocket in Fig. 4a is attributed to the upper branch of the surface state? Probably, the band structure of 1 QL films was not well understood in previous studies. New information would be appreciated.'

Authors' response:

If one ignores the energy range of the Dirac cone, the predicted mass gap of 1 QL Bi₂Se₃ based on our theory is about 0.84 eV. The predicted value is even larger than the energy gap of the bulk state at Γ (about 0.56 eV) [*Nat. Phys.* **5**, 438-442 (2009)]. The lower branch of surface state first drops into the bulk states because of the band shifting. In such case, the Dirac cone is no longer complete, and consequently, the calculated mass gap by our theory cannot be measured from ARPES for 1 QL.

We remeasured the ARPES spectrum of 1 QL Bi₂Se₃ in a larger energy range as shown in Fig. R1a (Fig. 4a in main text). There is only an electron pocket between the bulk valence band and the Fermi energy, and the band bottom is about -0.5 eV. As shown in Fig. R2 (Fig. 2 in the main text) and Fig. R3 (Fig. S3 in Supplemental information), with decreasing the thickness, the bulk conduction band evolves to several bound states and moves towards higher energy (all larger than -0.4 eV). Therefore, the electron pocket measured in 1 QL Bi₂Se₃ should be the upper branch of the surface state instead of the bound state, and the lower branch cannot be detected as expected.

Comment 3.2:

'If the authors utilize the data of 2.5 QL as a basis to exclude a phase transition between 2 QL and 3 QL, a more detailed explanation on it (and 1.5 QL) would be necessary. What state is 2.5 QL (or 1.5 QL film)? A mix of 2 QL and 3 QL islands?

'In relation, why are there more peaks in 2.5 QL film in Fig. S3 than in 2 QL or 3 QL? The band dispersion at 2.5 QL (and 1.5 QL) should be plotted somewhere. In addition, not only the mass but also the value of μ for 2.5 QL and 1.5 QL should be compared to the calculated one in Fig. 3.'

Authors' response:

Although the Van der Waals materials like Bi₂Se₃ grow layer by layer, the thickness of the film prepared by MEB is still difficult to be equal everywhere. As shown in Fig. R4 (Fig. S5. in Supplemental information), we measured AFM of 2.5 QL Bi₂Se₃. The height of islands and steps are integer QLs. Therefore, the APRES results should be a superposition of the results between two integer layers. The measurement of 1.5 and 2.5 QL is to improve the accuracy of the thickness-dependent evolution near the critical point.

The number of bound states decreases with decreasing the thickness. The changes of this number can be determined in ARPES spectrum directly. We have regulated the fitting process on the basis of bound states in the revised manuscript. As shown in Fig. R5 (Fig. S4 in supplemental information), now the numbers of Lorentzian peaks of 2, 2.5 and 3 QL samples are 3, 4 and 4, respectively.

The band dispersion curves for 2.5 QL and 1.5 QL were plotted in Fig. R3 (Fig. S3 in Supplemental information). Both the mass and the value of μ for 2.5 QL and 1.5 QL were compared to the calculated one and added in and R6 (Fig. 3 in main text).

Comment 3.3:

'Within this theory, is it generally difficult to achieve the QSH phase in ultra-thin films of any TI? If there are systems where QSH can still be realized even after taking into account Coulomb repulsion, can the authors predict what parameter region they are in?'

'In conclusion, the subject of this manuscript is important and would attract interests of the researchers in topological material sciences, but the points described above need to be addressed by the authors before I recommend publication.'

Authors' response:

Thank you for your recognition of the importance of our work. It is generally difficult to achieve the QSH phase in ultra-thin films by taking the inter-surface interaction into account. However, our work does not deny the previous work regarding the size effect on a QSH phase of the surface states. Therefore, there are still possibilities to find the QSH phase in other ultra-thin TI films.

If there are systems where QSH can still be realized even after taking Coulomb repulsion into account, the interaction-induced gap should be smaller than the size-

effect-induced one. In the case of Bi₂Se₃, the interaction-induced gap decays exponentially with the decay length of $\lambda/2 \approx 1.5$ nm (from Eq. (10) in main text), while the decay length of size-effect induced gap is about 50 nm [Eq. (12) in Phys. Rev. Lett. 101, 246807 (2008)]. This difference makes the expected region may appear in thicker samples. Another way to achieve this region is to adjust the ratio of Se vacancy to reduce the μ in Eq. 10.

Fig. R1.

Fig. R1. (Fig. 4 in main text). **ARPES spectra of the Bi₂Se₃(1 QL)/X/Si(111) system.** **a** is the ARPES spectrum of Bi₂Se₃(1 QL)/Si(111). X in **b** and **c** is Bi₂Te₃(2 QL) and Bi₂Te₃(1 QL)/Bi₂Se₃(1 QL), respectively.

Fig. R2.

Fig. R2 (Fig. 2 in main text). ARPES spectra of Bi₂Se₃/Si(111) samples with different thicknesses.

Fig. R3.

Fig. R3. (Fig. S3 in Supplemental information) **The ARPES spectra of Bi₂Se₃ with the thickness of half-integer thicknesses.** **a** and **c** are the ARPES spectra of 1.5 and 2.5 QL Bi₂Se₃ with their EDCs and Lorentzian fitting results shown in **b** and **d**, respectively.

Fig. R4

Fig. R4 (Fig. S5. in Supplemental information) **a** The AFM image of the Bi_2Se_3 film with the thickness of 2.5 QL. The scan size is $10 \times 10 \mu\text{m}^2$, and the oblique background is removed to show the steps. **b** The cross profile along the write line in **a**. The red dot lines in **b** are guide for the eye.

Fig. R5

Fig. R5 (Fig. S4 in supplemental information). **The fitting results of EDCs at Γ point with different thicknesses.** The black lines are the EDCs of the ARPES spectra, while the orange lines and red lines are the Lorentzian fitting of the single peak and the total EDCs.

Fig. R6

Fig. R6 (Fig. 3 in main text). **The results of Theoretical Validation.** a and b are the experimental and theoretical results of the Dirac position μ and the magnitude of the gap $2m$, respectively. The cyan circles in b are the results of Y. Zhang, *et al.*

REVIEWERS' COMMENTS

Reviewer #3 (Remarks to the Author):

The authors have addressed all the issues raised in my report. Therefore, I recommend the publication.

PS, the following typos should be corrected.

p. 8, line 158, MEB

p. 8, line 159, APRES

RESPONSE TO REVIEWER COMMENTS

Response to Reviewer #3:

Overall comments:

'The authors have addressed all the issues raised in my report. Therefore, I recommend the publication.

'PS, the following typos should be corrected.

'p. 8, line 158, MEB

'p. 8, line 159, APRES'

Authors' response:

Thank you for your careful reading and your recommendation of our work. The typos are fixed in revised manuscript.